# Enhancement of Molecular Transport into Film Stacked Structures for Micro-Immunoassay by Unsteady Rotation

**DOI:** 10.3390/mi14040744

**Published:** 2023-03-28

**Authors:** Hinata Maeno, Satoshi Ogata, Tetsuhide Shimizu, Ming Yang

**Affiliations:** Department of Mechanical System Engineering, Faculty of System Design, Tokyo Metropolitan University, Tokyo 191-0065, Japan

**Keywords:** enzyme linked immunosorbent assay (ELISA), microfluidics, unsteady-rotation

## Abstract

A film-stacked structure consisting of polyethylene terephthalate (PET) films stacked in a gap of 20 µm that can be combined with 96-well microplates used in biochemical analysis has been developed by the authors. When this structure is inserted into a well and rotated, convection flow is generated in the narrow gaps between the films to enhance the chemical/bio reaction between the molecules. However, since the main component of the flow is a swirling flow, only a part of the solution circulates into the gaps, and reaction efficiency is not achieved as designed. In this study, an unsteady rotation is applied to promote the analyte transport into the gaps using the secondary flow generated on the surface of the rotating disk. Finite element analysis is used to evaluate the changes in flow and concentration distribution for each rotation operation and to optimize the rotation conditions. In addition, the molecular binding ratio for each rotation condition is evaluated. It is shown that the unsteady rotation accelerates the binding reaction of proteins in an ELISA (Enzyme Linked Immunosorbent Assay), a type of immunoassay.

## 1. Introduction

Immunoassay is a biochemical test commonly used in hospitals and laboratories. Since its development in the 1950s, it has been used for tests from home pregnancy to AIDS tests to recent COVID-19 diagnoses [1,2,3]. Immunoassays are classified into multiple types according to label form, such as enzymes, radioisotopes, fluorescent dyes, chemiluminescent probes, etc. Enzyme-based labels, easy to molecularly design for the entire assay, are the most used, the so-called ELISA (Enzyme-linked immunosorbent assay). ELISA immobilizes a primary antibody on the surface of a polystyrene 96-well plate to detect a specific antigen in the sample. Although this method has high specificity, sensitivity, and the ability to perform multiple assays simultaneously, antigen—antibody binding takes a long time because antigen diffusion depends on Brownian motion [4,5]. Additionally, microfluidic devices are collectively used to create microfluidic channels and reaction vessels using microfabrication technologies, such as MEMS technology, and apply them to biotechnology and chemical engineering. ELISA is expected to shorten the analysis time and increase sensitivity by shortening the diffusion distance to the captured antibody and expanding the specific surface area of the reaction field. Various microfluidic devices, such as channel patterning [6], microbeads [7], centrifuge disks [8], and paper-based devices [9,10], have been developed for ELISA. In ELISAs using microbeads, a captured antibody coating is applied to the surface of the beads and placed in a sample solution to perform the reaction. In addition to obtaining a large surface area by using multiple beads, the time it takes for the beads with the captured antibody to bind to the molecules in the sample is shortened by stirring [11]. The centrifugal disk creates a microfluidic channel on the disk and transports the solution by centrifugal force. The reaction chamber is made small, so the sample volume is also small, and as a result, the reaction time shortens. In addition, the separation of serum from the blood, which is usually done before ELISA as a sample preparation, centrifugal force has also been developed to perform in devices [12,13]. Although various microfluidic devices have been developed, most laboratories still use the conventional 96-well plate for the assays [14]. This is due to the high cost of device fabrication [15] and the need for peripheral equipment, such as pumps and detectors [14]. In medical and biological applications, disposable devices are desirable to avoid biological contamination and false-positive signals; therefore, low-cost and mass-producible microfluidic devices are required [16]. However, the soft lithography technique, the most common microfluidics fabrication process, is costly because it requires photolithography for mold fabrication [15]. Even if the manufacturing cost of the device can be reduced, the conventional plate reader cannot be used, and a detector needs to be developed for quantification.

To address these concerns, the authors have developed a low-cost microfluidic device called the 3D-Stack that can be combined with 96-well plates [17]. The 3D-Stack is made by stacking polyethylene terephthalate (PET) films, with 20 µm gaps between the films, to serve as microfluidic paths to increase the binding area by using the surface of the films. The entire assembly process of 3D-Stack is performed using a press machine, which facilitates mass production and lowers the cost of device production. The entire 3D-Stack is coated with capture antibodies and rotated in a 96-well plate to provide circulating flow in the gaps to increase antigen binding. For subsequent reaction steps, the 3D-Stack can be easily moved to the adjacent well containing the reagents; finally, to quantify the enzyme reaction, the 3D-Stack is removed, and the remaining reaction solution in the wells can be read using a conventional plate reader. In our previous studies, the detection sensitivity of the 3D-Stack was higher than that of conventional methods when measurements were performed with the same reaction time as conventional methods [18,19]. Suzuki focused on the flow between films and suggested that there is a trade-off between surface area increase and flow rate increase, which affects the sensitivity of ELISA [20]. It was shown that surface modification improves the density of captured antibodies and can detect IgA (a protein contained in human saliva) in less than half the measurement time of conventional methods [21]. Detection with the 3D-Stack can quantify CD163 from dengue patient sera with the same accuracy as the conventional ELISA [22]. However, the reaction time of the 3D-Stack was longer than that of microfluidic channels with the same channel dimensions. The 3D-Stack requires about 30 min for incubation with a channel height of 20 µm, while the microfluidic channel of Weng et al. requires 3–7 min for incubation with a channel height of 80 µm [23]. In addition, Nakajima et al. reported that the enzymatic reaction time was only a few seconds in a microchannel with a channel height of 20 µm [24].

One possible reason for this is that the solution in the wells did not circulate uniformly, and the analyte was not transported into the microfluidic in the gaps of the 3D-Stack. When the 3D-Stack is in steady rotation, the surrounding solution is also in steady rotation. In this case, the centrifugal force and pressure gradient on the fluid elements are balanced in the rotational coordinate system. Therefore, some analytes will continue to rotate on the same circumference and stay in the corner of the well.

A similar problem occurs in laminar mixing using stirred tanks. When mixing under low Reynolds number conditions, two regions are formed in the mixing tank: the active mixing regions (AMR) and the isolated mixing region (IMR), which are toroidal vortices above and below an impeller [25]. In AMR, mixing occurs by convection, whereas in IMR, the material exchange is performed only by diffusion at the interface between AMR and IMR, which inhibits uniform mixing in the entire stirred tank. Therefore, many researchers have developed different ways of destroying the IMR. The placement of baffles enhances the radial momentum exchange between the near-wall and the bulk fluid, and IMR is not generated [26,27]. Eccentricity also improves axial mixing by disrupting spatial symmetries [28,29]. However, these attempts are costly because they require additional structures or complex rotating mechanisms. Another effort to break IMR is unsteady rotation, such as forward-reverse mixing and time-dependent rotational speed [30,31]. However, these studies were conducted using impellers with strong shear force action and discharge force, and it is difficult to achieve the same effect by rotating a bladeless disk such as a 3D-Stack in the same way. In this study, we made a new attempt to break the IMR using the secondary flow generated on the surface of the rotating disk. When the disk is rotating steadily, the centrifugal force and the centripetal force on the fluid element are balanced. However, when the disk is suddenly stopped, the fluid rotating near the disk surface is subjected to viscous resistance, and the centrifugal force decreases. As a result, the centrifugal force due to the pressure gradient generates a secondary flow in the radial direction [32]. We aim to use this secondary flow to transport the analyte in the IMR to the inlet of the 3D-Stack. Currently, rapid influenza, COVID-19, and pregnancy tests are rapid. The immunochromatography method can be detected in about 15 min, but it is difficult to increase sensitivity, and it is limited to simple qualitative analysis. Therefore, ELISA is used when detecting trace substances or when quantitative analysis of the target molecule is required in disease diagnosis, although it takes a long time to measure and cannot respond to rapid diagnosis. We attempted to shorten the measurement time of ELISA to 15 min, which is the same as a simple test. To achieve this, it is necessary to perform each ELISA reaction step within 5 min; but even if the rotation of the 3D-stack is up to 3000 rpm, the same sensitivity as the conventional method cannot be obtained unless each reaction step is given at least 10 min. The cause of this may be that the circulation flow of the solution is uneven, causing molecule retention. However, the ideal circulation state has just been assumed, and the circulation state of the solution and the molecular transport to the flow path have not been verified yet.

The purpose of this study is to investigate the effect of unsteady rotation on the flow and concentration distribution in the well by stopping and re-rotating the 3D-Stack. A finite element analysis is performed to simulate flows under steady and unsteady rotations. The changes in flow and concentration distribution for each rotation operation are evaluated. Furthermore, the simulation results are adopted to optimize the rotation conditions for the ELISA process. In addition, the molecular binding ratio is evaluated for each rotation condition. The optimal condition for the unsteady rotation is applied to an ELISA using Human IgA to evaluate its efficiency experimentally.

## 2. Materials and Methods

### 2.1. Materials and Instruments

Human IgA ELISA Quantitation kit (E80-102), ELISA Blocking Buffer (E104), Wash Solution(E106), and TMB One Component Substrate (E102) were purchased from Bethyl Laboratories. Sodium hydroxide (NaOH), 2-(N-morpholino)ethanesulfonic acid (MES), N-(3-Dimethylaminopropyl)-N′-ethylcarbodiimide hydrochloride (EDC), and N-Hydroxysuccinimide (NHS) were acquired from Sigma-Aldrich (St. Louis, MI, USA). TMB stop solutions were purchased from SeraCare Life Sciences Inc (Milford, MA, USA). All chemicals and reagents were of analytical grade and used without further purification. A 96 well microplate (MS-8496F) was purchased from Sumitomo Bakelite Co., Ltd (Tokyo, Japan). The absorbance was measured by a Microplate reader Infinite^®^F50 (Tecan Group Ltd., Mannendorf, Switzerland). The 3D-Stacks were fabricated by Polyester Film Lumirror™ (Toray Industries, Inc., Tokyo, Japan). The detailed fabrication method was described in a previous study. Figure 1 shows a schematic image for fabricating a 3D-Stack and utilization within a well. The DC motor (DMN37BA, Nidec Servo Corporation, Kiryu, Japan) for the rotating 3D-Stack was controlled by Arduino uno Rev3 and motor driver (TB6643KQ, TOSHIBA, Inc., Tokyo, Japan).

### 2.2. Mathematical Models and Numerical Method

A multi-physis simulation for microflow, diffusion, and molecule binding was performed for a well with 3D stacks in steady and unsteady rotations. The mathematical models are shown below.

#### 2.2.1. Flow in Well

The analyte in a well with a rotating 3D-Stack moves toward the rotational and circumferential directions due to the rotation of the 3D-Stack and creates a flow. The flow is assumed to be laminar and follows the Navier-Stocks equation, as shown in Equation (1).
(1)$$\rho \left\{\frac{\partial v}{\partial t}+(v\cdot \nabla )v\right\}=-\nabla p+\mu \nabla^{2}v+\rho f$$
∇⋅v=0
where ***v*** is the velocity of the flow, *μ* the flow velocity, and *ρ* the density.

#### 2.2.2. Bulk Analyte Transport

The transport of the antigen in the bulk liquid phase is described by the convection–diffusion equation,
(2)$$\frac{\partial C}{\partial t}=\nabla \cdot \left(D\nabla C\right)-\nabla \cdot \left(vC\right)$$
where *C* is the concentration of the antigen in the bulk phase, *D* is the diffusion coefficient of the antigen, and ***v*** is the flow velocity.

#### 2.2.3. Surface Binding Kinetics

The capture-antibody is immobilized on the surface of the 3D-Stack. The binding reaction with the analyte at the surface is described by,
(3)$$v=\frac{d\left[C_{s}\right]}{dt}=D_{s}\nabla^{2}\left[C\right]+k_{on}(\left[C_{0}\right]-[C_{s}])(\left[B_{0}\right]-[C_{s}])-k_{off}[C_{s}]$$
where *B* is the concentration of the free binding sites at the surface and *C_S_* is the concentration of the antigen–antibody complexes at the surface. *k_on_* and *k_off_* are association rate and dissociation rate constants, respectively.

#### 2.2.4. Computational Environment and Analytical Model

COMSOL Multiphysics ver5.6 (COMSOL AB Inc., Stockholm, Sweden) was used for the numerical analysis software. A Precision T3600 (DELL Inc., Round Rock, TX, USA) with an Intel(R) Xeon(R) E5-1660 CPU (6 cores, 3.3 GHz) and 4 slots of 8 GB DDR3 memory was used for calculations.

A rotationally axisymmetric two-dimensional model of the 3D-Stack in a well containing 100 µL of solution was utilized. The 3D-Stack was designed to have an outer diameter of 5 mm, an inner diameter of 2 mm, a film thickness of 0.1 mm, and a film spacing of 20 µm. The specification of a 3D-stack in a well and its simulation model are shown in Figure 2. Triangular elements were used for the mesh, and two layers of boundary layer mesh were created for the container and the structure walls. The boundary layer mesh was used to accurately capture changes in physical quantities, such as concentration and velocity gradients, near the wall surface. The number of elements was 40,797.

### 2.3. Calculation of Interfacial Area between High and Low Concentration Area

To discuss the necessity of introducing unsteady rotation, the calculation of concentration distribution and the interface between the high and low concentration areas was performed. An imaging process was applied to evaluate the interfacial area. Figure 3 shows a typical image of the concentration area for the analyte concentration of higher than 90% via binarization.

### 2.4. Sandwich ELISA Using a 3D-Stack and Conventional Method

For an ELISA using the 3D-Stack, goat anti-human IgA antibody was diluted to 10 µg/mL with 0.05 M CBS, pH 9.6, and added to a 96-well plate at 100 µL per well. One 3D-Stack was inserted into each well and rotated by a motor stirrer at 2000 rpm for 5 min at room temperature. After incubation, the 3D-Stack was transferred to the well, containing 100 µL of TBS-T (50 mM Tris, 0.14 M NaCl, 0.05% Tween20, pH 8.0) as washing solution, and rotated at 2000 rpm for 10 s, 5 times. For blocking, 100 µL of 1% BSA in TBS (50 mM Tris, 0.14 M NaCl, 0.05%, pH 8.0) was added to a new well. The 3D-Stack was inserted and rotated for 5 min at room temperature. After blocking, purified Human IgA was diluted with 1% BSA in TBS-T and added to each new well. The 3D-Stack was inserted into the well and rotated at 2000 rpm for 5 min, followed by washing. Next, 100 µL of the detection antibody (HRP conjugated goat anti-human IgA antibody), diluted to 1 µg/mL with 1% BSA in TBS-T, was added to each new well. The 3D-Stack was inserted and rotated at 2000 rpm for 5 min at room temperature, followed by washing. Lastly, 100 µL of TMB substrate was added to each new well. The 3D-Stack was inserted and rotated at 2000 rpm for 5 min at room temperature in the dark, followed by washing. After the reaction, the 3D-Stack was removed from the well and the absorbance of the remaining solution in the well was measured at 450 nm wavelength, with a reference wavelength of 570 nm, using a microplate reader. The reagents used were based on previous studies [17]. The process conditions, including the duration of each step in the unsteady rotation, were based on the simulation results described in the next section. For conventional ELISA using only 96-well plates, the same concentrations of reagents were used, and the assay was performed according to the protocol provided by the manufacturer (Human IgA ELISA Quantitation Set, catalog no. E80-102, Bethyl Laboratories, Inc., Waltham, MA, USA). Briefly, the incubation times were as follows: 1 h for capture antibody, 30 min for blocking, 1 h for sample, 1 h for detection antibody, and 15 min for TMB substrate. Corrected absorbance values were obtained after subtracting the absorbance of blank wells. The scheme of a sandwich ELISA using a conventional 96-well plate and 3D-Stack methods is shown in Figure 4.

### 2.5. Evaluation of Binding Rate

3D-Stack was rotated for 1 to 5 min in a solution containing 10 ng/mL of human IgA. The concentration of unbound human IgA remaining in the solution was determined by sandwich ELISA using 96 wells. The binding rate to the 3D-Stack was calculated by subtracting the concentration of unbound human IgA from the initial concentration (Figure 5).

### 2.6. Evaluation of Detection Limit

To evaluate the performance of a 3D-Stack using unsteady rotation, detection, and quantification, limits were calculated and compared to the conventional ELISA method using 96 wells. Limits of detection and quantitation were calculated from the slope of the calibration curve and the standard deviation of the blank value based on the definition of ICH (International Council for Harmonisation of Technical Requirements for Pharmaceuticals for Human Use) [33]. Each standard sample was repeated twice for duplicate measurements. The coefficient of variation in the measured values of the standard samples was also calculated to evaluate the measurement variability.

## 3. Result and Discussions

### 3.1. Concentration Distribution with Steady Rotation

The changes in the concentration distribution of the analytes from the start of the reaction to 5 min later in the steady rotation condition were calculated, and the results are shown in Figure 6. The concentration distribution is shown in the grayscale, where the white is closer to the initial concentration and the black is the concentration of a zero percentage of analyte in the solution, decreased due to the analyte binding to the capture antibody on the 3D-Stack surface. This stagnant area, which concentration is close to the initial value, still exists after 5 min from the start of the reaction. Since the 3D-stack has axis symmetry, the concentration distribution in the rotation direction is almost the same. The reason why diffusion does not proceed sufficiently even though the wells are polarized into a high concentration area (white) and a low concentration area (black) is that the protein to be analyzed is less diffusible than small molecules such as ions and the diffusion force due to the concentration gradient is small because the initial concentration is low in the trace analysis.

The percentage of the analytes in the solution bound to the 3D-Stack was also calculated and evaluated as the binding rate. The results for each reaction time are plotted in Figure 7. This graph shows that the binding rate increased rapidly within one minute after the start of the reaction and then changed gently after that. This is probably because only a part of the solution was circulating in the gaps of the 3D-Stack, and the analytes in the rest of the solution were transported only by diffusion. The binding rate after 5 min of reaction was 47.9%.

### 3.2. Concentration Distribution with Unsteady Rotation

The concentration distributions before and after stopping the flow are also calculated, and the results are shown in Figure 8. A radial inward flow occurs on the surface of the 3D-Stack after shutdown, and the molecules in the corner of the well move toward the inlet, meaning that the stop motion breaks the balance of the centrifugal force and the centripetal force on the fluid element via rotation. As a result, a secondary flow is generated in the radial direction, which enhances the analyte in high-concentration areas and diffuses to low-concentration areas.

Figure 9 shows the calculated concentration distribution after stopping at 0.1 s and restarting the rotation. The concentration distribution after re-rotation is more uniform than before stopping, and some of the moving molecules flow into the gaps of the 3D-Stack. This is because the highly concentrated fluid masses in which the molecules were stagnant are stretched by the flow generated via the stopping and re-rotating operation. Microscale mixing is done by diffusion [34]. Therefore, when mixing solutions A and B, increasing the interfacial area of the solutions promotes diffusion. In the case of a microfluidic mixer, the interfacial area is increased by increasing the number of branches in the flow path [35]. In addition, J. Baldya et al. explained the mechanism of micro-mixing as an engulfment deformation diffusion (EDD) model, in which the fluid mass is entrained in the vortex and stretched, leading to molecular diffusion [36]. The results of the present analysis can also be interpreted as the high-concentration fluid masses formed above and below the 3D-Stack during the steady rotation. They are stretched by the radial inward flow due to stopping, and the vortex structure generated by the re-rotation promotes molecular diffusion by increasing the interfacial area.

The calculated molecular binding rates during repeated stopping and rotating operations compared to the case for steady rotation are shown in Figure 10. The variation of the molecular binding rate is larger than that of the steady rotation. This means that the unsteady rotation promoted the diffusion of the analyte in the solution, and the higher volume of the analyte circulated into the gaps of the 3D-Stack. As a result, the binding rate of an analyte with immobilized antibodies increased, i.e., the reaction efficiency in chemical/bio reactions can be promoted significantly.

### 3.3. Optimize Unsteady Rotation

Unsteady rotation promotes analyte dispersion by expanding the interfacial area between the high and low-concentration regions. For optimizing the process conditions, the stopping time required to increase the interfacial area and re-rotation time after the stop must be determined. The interfacial area changes after stopping the 3D-Stack from 2000 rpm with a deceleration time of 0.1 s from 0.1 s to 0.5 s were calculated. The results are shown in Figure 11. The figure shows that the interfacial area increases rapidly at 0.1 s after stopping and changes little after that. From this, it can be said that adding a 0.1 s stop motion between rotations effectively disperses retained analytes.

Next, the molecular binding rate was numerically evaluated after repeating the stopping and re-rotating for 30 s by stopping time in 0.1 s and changing rotation time from 0.1 s to 10 s. The results are shown in Figure 12. The figure shows that the binding rate reached its maximum value when the rotation time was 1.0–1.5 s. To investigate this factor, the behavior of the solution reacting with the captured antibodies on the 3D-Stack surface was evaluated. The solution whose concentration decreased from the initial value is defined as the reacting solution, and its volume is plotted against time in Figure 13. The volume of the reaction solution increased rapidly in the first second and changed little thereafter. This is because the reaction solution circulates the 3D-Stack in about 1 s, and after that, the analyte moves to the 3D-Stack via diffusion. Therefore, the optimal rotation time is when the solution circulates around the 3D-Stack by convection. In this study, the optimal condition for unsteady rotation is repeating rotation in 1 s and stopping in 0.1 s. However, in this case, the flow rate between the stacked films is much smaller than the flow around the cylinder since the gap between the layers is only 20 mm. For the condition of a rotation speed of 2000 rpm, the fluid Reynolds number is about 0.05. The gap distance could influence the reaction efficiency, and optimal gap distance needs to be discussed in future work.

### 3.4. The Effect of Unsteady Rotation on Binding Rate

The binding rates for each reaction time are evaluated experimentally, and the results are shown in Figure 14. In the case of steady rotation, the binding rate after 5 min was 25.9%, while in the case of unsteady rotation, the binding rate increased to 84.9%. This means the unsteady rotation significantly increases the amount of molecular binding per unit of time.

### 3.5. The Effect of Unsteady Rotation on Detection Limit

ELISA using IgA with various concentrations was executed to obtain the calibration using a 96 well with and without the 3D-Stack. The results are shown in Figure 15. Both axes of the figure are logarithmic. The absorbance for the 3D-Stack is higher than that of conventional ELISA for all samples. Table 1 shows the limits of detection and quantification calculations based on this calibration curve. The coefficients of variation in the absorbance result for each condition of ELISA are shown in Figure 16. The coefficients of variation for both conventional and 3D stacks are lower than 10%. From the experimental results, ELISA with a 3D-Stack reduced the measurement time by a factor of 9 and improved the detection limit by more than 60 times. It is seen that the unsteady rotation is very effective in promoting diffusion and enhancing the analyte circulation. As a result, the measure time and detection limit are improved significantly. We think that the 3D-Stack with a conventional 96 well plate could be a novel device for rapid assays, which has a high potential for time-saving and higher cost performance than other microdevices for rapid assays.

## 4. Conclusions

The effect of unsteady rotation on the flow and concentration distribution in the well by stopping and re-rotating the 3D-Stack was investigated numerically and experimentally. A finite element analysis based on multiphysics analysis was performed to simulate flows under steady and unsteady rotations. An optimal condition for the unsteady rotation was obtained numerically, and the efficiency was evaluated by an ELISA experimentally. The conclusions and findings are summarized as follows:(1)Unsteady rotation with repeated stops and re-rotation promotes the diffusion of the analyte in the solution and enhances circulation into the gaps of the 3D-Stack while the analyte remains in the corners of the wells for steady rotation;(2)The unsteady rotation increased the molecular binding rate by about 60% in a 5-min antibody-antigen reaction compared to steady rotation;(3)Non-steady rotation reduced the measurement time to less than one-ninth of the conventional ELISA and improved the detection limit by more than 60-fold. The value of the coefficient of variation was less than 10%, and there was no effect on variation.

## Figures and Tables

**Figure 1 micromachines-14-00744-f001:**
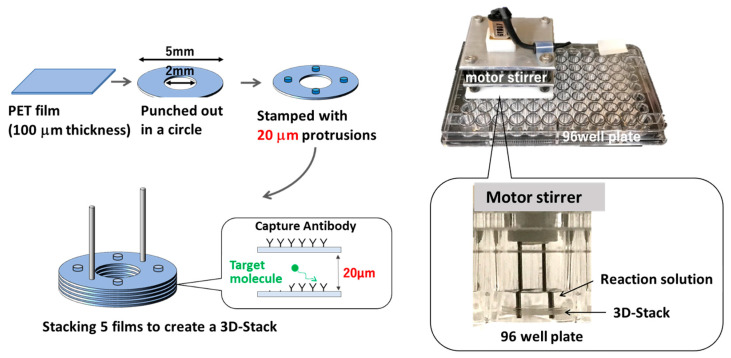
Schematic image for fabrication of 3D-Stack and utilization within a well.

**Figure 2 micromachines-14-00744-f002:**
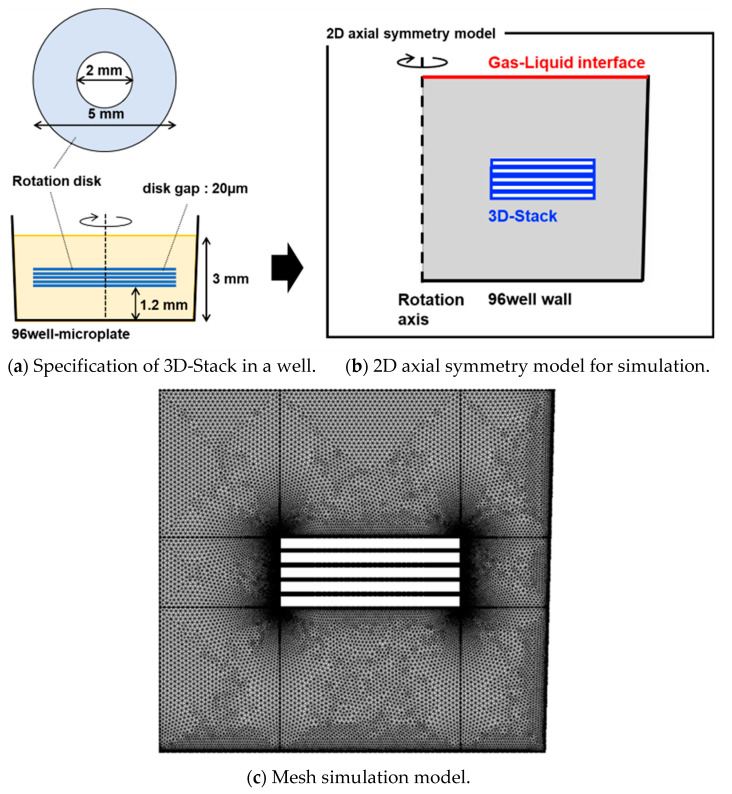
Specification of 3D-Stack in a well and the simulation model.

**Figure 3 micromachines-14-00744-f003:**
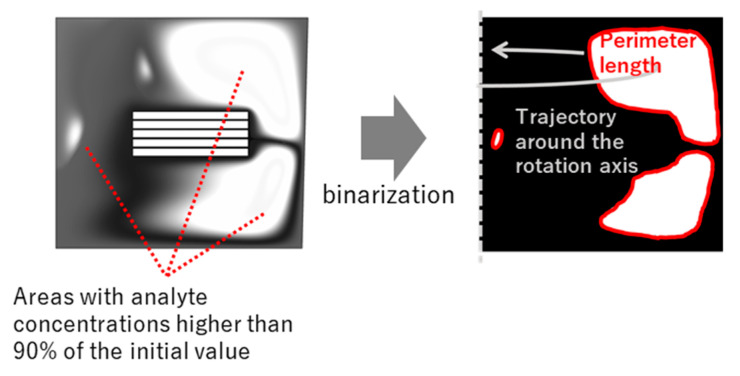
An imaging process for calculation of high concentration area.

**Figure 4 micromachines-14-00744-f004:**
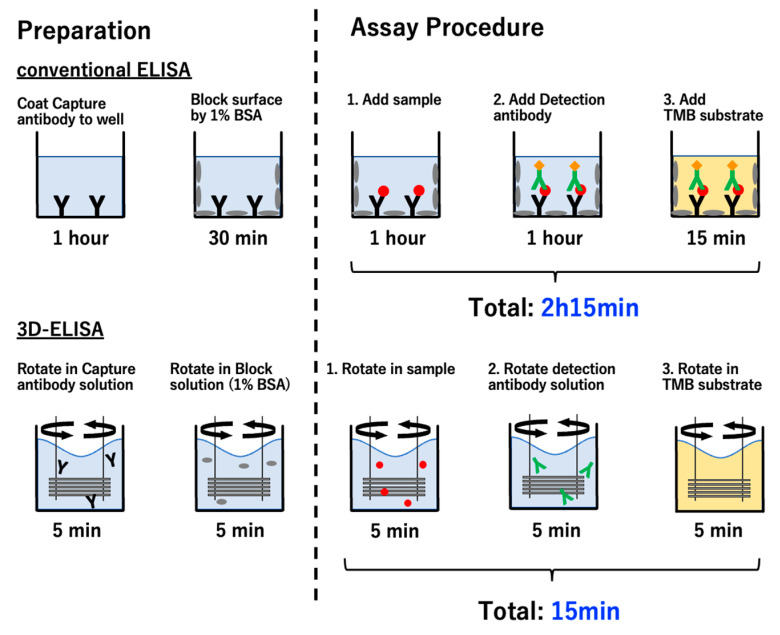
Scheme of sandwich ELISA using a conventional 96-well plate and 3D-Stack methods. Black Y is goat anti-human IgA antibody; Green Y with orrange dots is detection antibody with HRP; gray dots are BSA for blocking; red dot are human IgA, Orange squares in last step are AMB substrate.

**Figure 5 micromachines-14-00744-f005:**
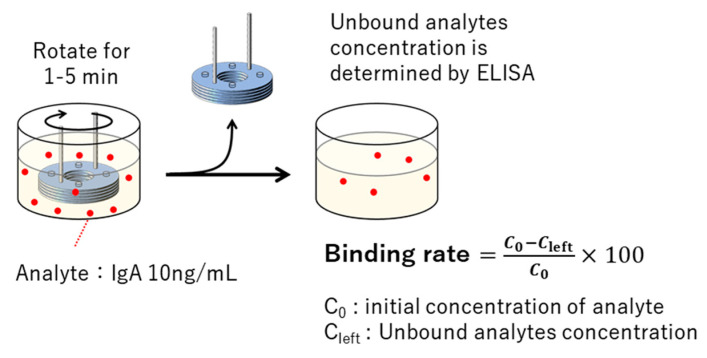
Process for evaluation of binding rate in a sandwich ELISA.

**Figure 6 micromachines-14-00744-f006:**
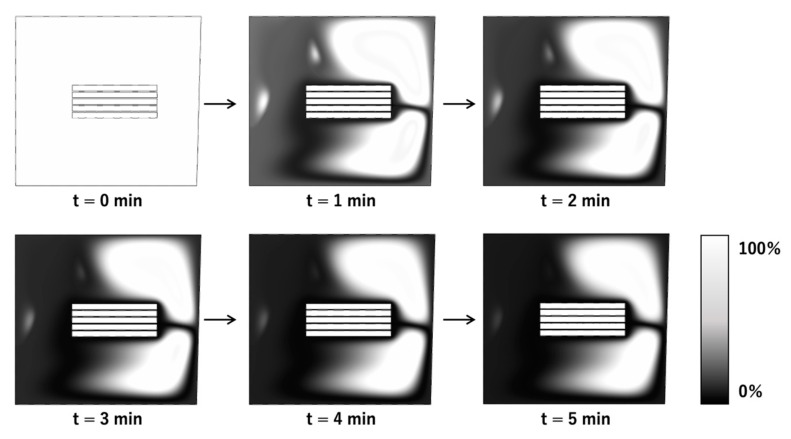
Calculated changes in concentration distribution of analytes in steady rotation conditions.

**Figure 7 micromachines-14-00744-f007:**
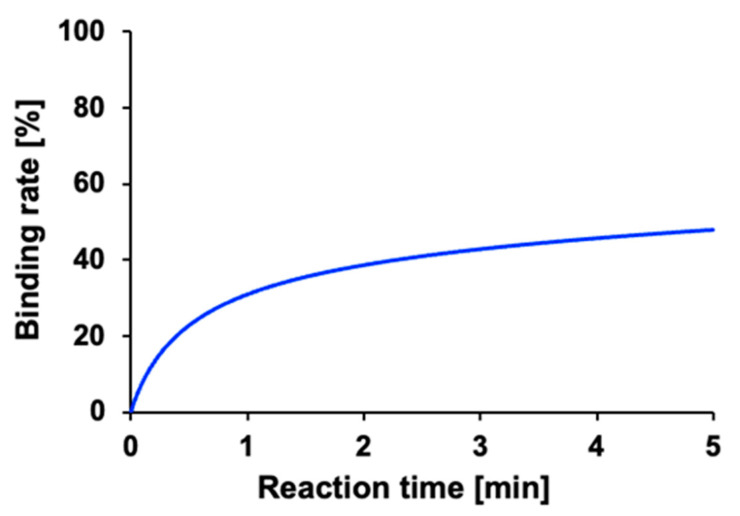
Calculated increase in binding rate during steady rotation.

**Figure 8 micromachines-14-00744-f008:**
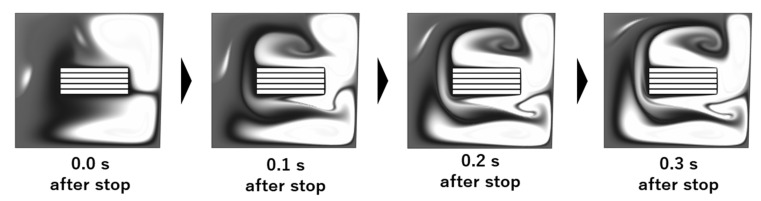
Calculated image of variation in concentration distributions after stopping the flow.

**Figure 9 micromachines-14-00744-f009:**
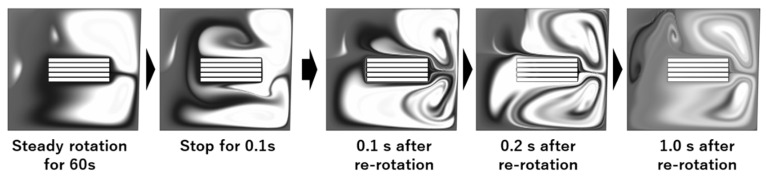
Calculated variation in concentration distributions after stopping the flow in 0.1 s and re-rotation.

**Figure 10 micromachines-14-00744-f010:**
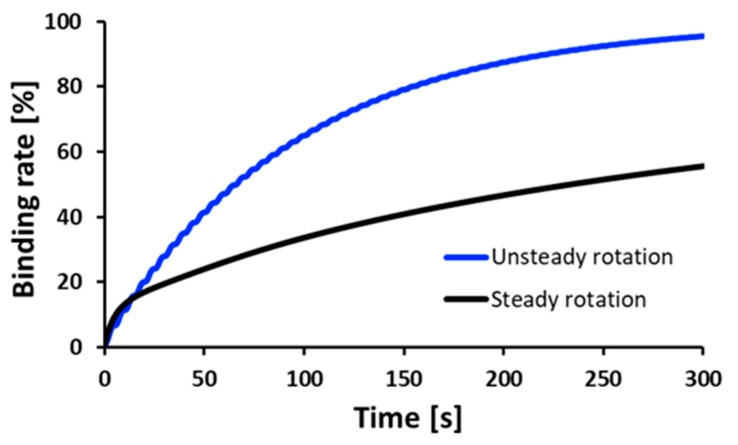
Calculated increase in the binding rate for stop and re-rotation compared to steady rotation conditions.

**Figure 11 micromachines-14-00744-f011:**
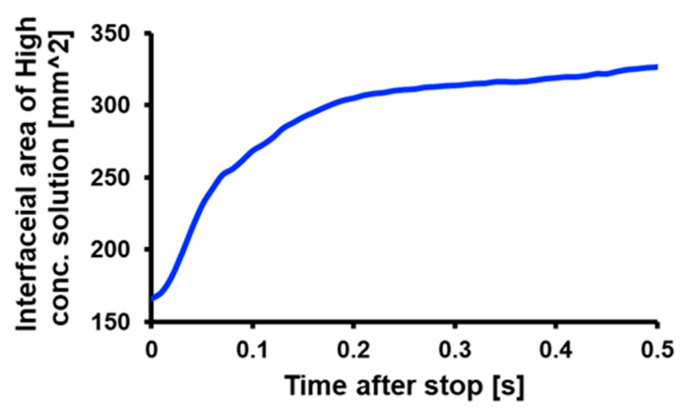
Calculated variation in interfacial area between high and low concentration.

**Figure 12 micromachines-14-00744-f012:**
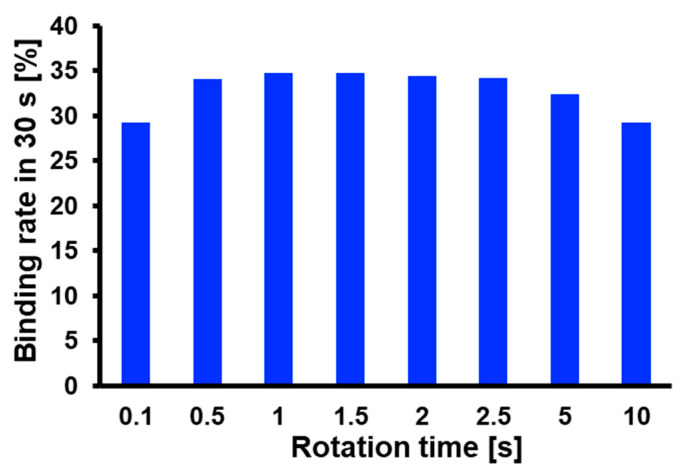
Calculated variation in binding rate after repeating the stopping and re-rotating for 30 s by stopping time in 0.1 s and changing rotation time from 0.1 s to 10 s.

**Figure 13 micromachines-14-00744-f013:**
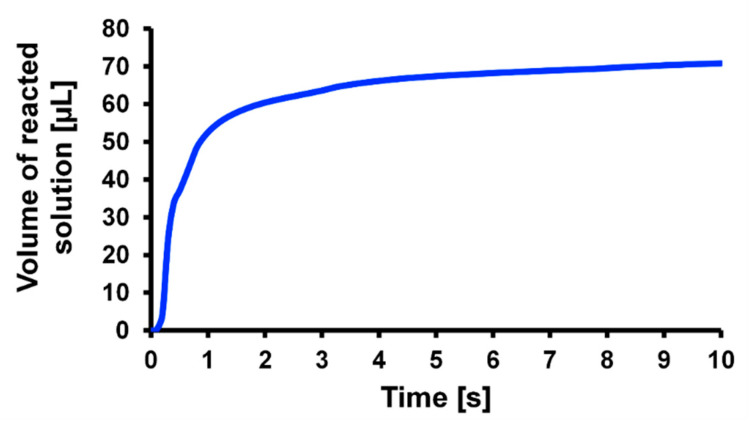
Calculated variation in volume of reacted solution responding to rotation time.

**Figure 14 micromachines-14-00744-f014:**
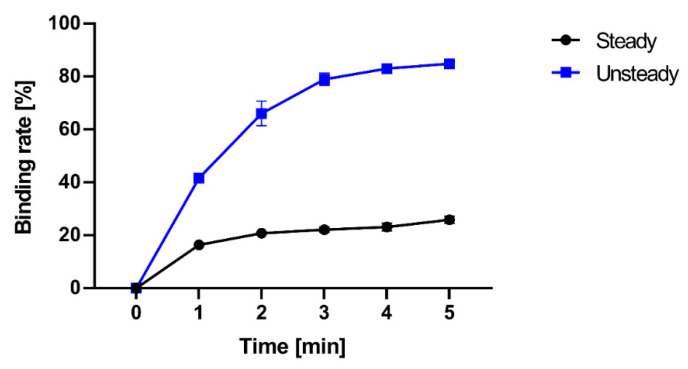
Experimental results on binding rate for various reaction times.

**Figure 15 micromachines-14-00744-f015:**
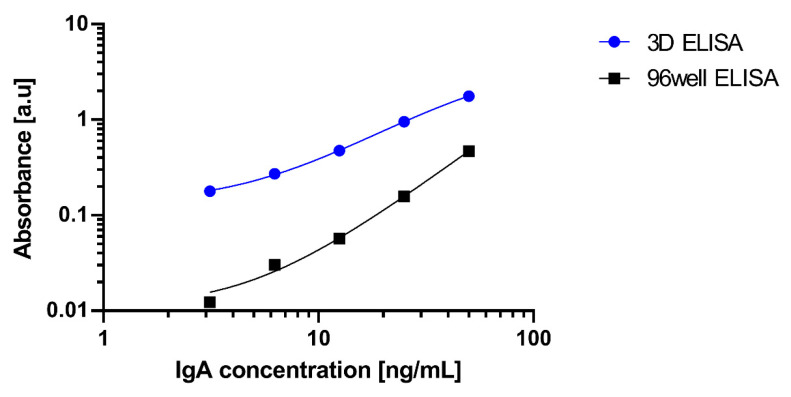
Calibration curve for ELISA of IgA using 96 well and 96 well with 3D-Stack.

**Figure 16 micromachines-14-00744-f016:**
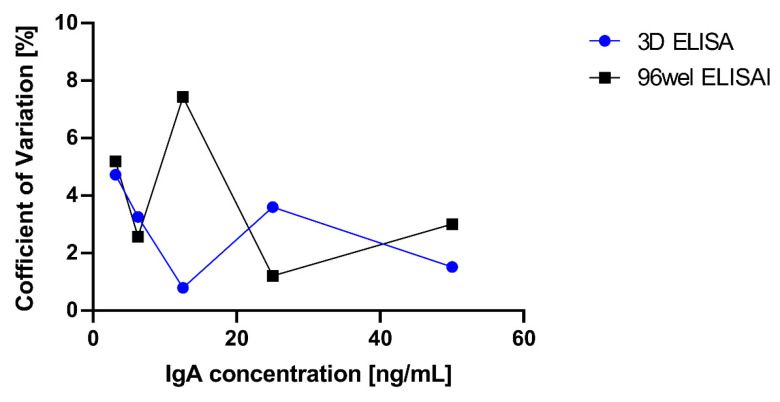
Coefficient of variation in the result of absorbance for each condition of ELISA.

**Table 1 micromachines-14-00744-t001:** Comparison in detection and quantification limits and process time for ELISA.

	Detection Limit (ng/mL)	Quantification Limit (ng/mL)	Turn Around Time (min)
Conv.	20.7	53.2	135
3D-Stack	0.32	4.79	15

## Data Availability

The data presented in this study are available upon request from the corresponding author.

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
