# Peer review of "Enhancement of Molecular Transport into Film Stacked Structures for Micro-Immunoassay by Unsteady Rotation"

_micromachines, 2023, doi:10.3390/mi14040744_

Round 1

Author Response

Dear reviewer,

Thanks for reviewing our paper.

Please found answers in attached file to your questions.

Reviewer 2 Report

This manuscript tried to enhance the transport in film stacked structures. Their idea is to apply unsteady rotation to destroy the isolated mixing region. The authors conducted experiments and numerical simulations to ensure the feasibility of the approach. Generally, the manuscript’s novelty lies in the idea of the unsteady rotation. However, the writing of the manuscript is far below the standard of scientific precision. I think it can be published after the authors significantly improve the manuscript to clarify some issues in their study. I listed some suggestions, and I hope the authors check the manuscript carefully to ensure it is self-consistent.

1. Authors should clearly demonstrate the setup of numerical details, such as boundary condition and grid in the simulation. Did they check grid independence? In Fig.2, what is the boundary condition at the gas-liquid interface? Which method did they use in the interface tracking? How many grids did they use to capture the 20um film space? What is the numerical scheme used?

2. How to verify the reliability of the computation? Is there any comparison between experiments and simulations to ensure the accuracy of the computations?

3. In 2.3, the authors try to demonstrate how they calculate the reaction area. However, the demonstration is too vague for a quantitative calculation. At least they should present a reference for the binary method.

4. The manuscript is vague when they used experimental results or numerical results. In every figure, the manuscript should clearly state the working conditions and parameters.

5. The data of Table 1 is already illustrated in Figure, so it is unnecessary. For example, in Fig. 13, the author didn’t demonstrate what the parameters for the unsteady rotation.

6. In line 418, “In many of the examples, the article does not clearly state the working conditions and parameters.” The expression is quite vague for readers to understand.

7. The present tense and the past tense should be consistent in the manuscript.

Author Response

Dear Reviewer,

Thanks for reviewing our paper.

Please found answers in attached file to your questions.
